# Stochastic switching and analog-state programmable memristor and its utilization for homomorphic encryption hardware

Woon Hyung Cheong [1], Jae Hyun In [1], Jae Bum Jeon[1], Geunyoung Kim[1] & Kyung Min Kim [1] ✉

Homomorphic encryption performs computations on encrypted data without decrypting, thereby eliminating security issues during the data communication between clouds and edges. As a result, there is a growing need for homomorphic encryption hardware (HE-HW) for the edges, where low power consumption and a compact form factor are desired. Here, a $Pt/Ta_2O_5/Mo$ metallic cluster-type memristors (Mo-MCM) characterized by the Mo as a mobile species, and its utilization for the HE-HW via a 1-trasistor-1-memristor (1T1M) array as a prototype HE-HW is proposed. The Mo-MCM exhibits inherent stochastic set-switching behavior, which can be utilized for generating the random numbers required for encryption key generation. Furthermore, the device can accurately store analog conductance states after set-switching, which can be used as an analog non-volatile memristor. By simultaneously leveraging these two characteristics, encryption key generation, data encryption, and decryption are possible within a single device through an in-memory computing manner.

In light of the latest advancements in social networking and IoT technology, a massive amount of data is generated and transmitted in our vicinity[1–3]. The surge in data demands poses challenges for conventional Boolean logic-based computing approaches, particularly in terms of energy consumption. Consequently, there is a need for computing capabilities to address these challenges effectively via the development of specialized and personalized hardware tailored for more energy-efficient execution of computing tasks. These emerging hardware directly utilize physical phenomena to implement specific computations. In this regard, memristors are highly versatile, as they can provide various physical behaviors useful for computation in a simple capacitor-like structure. For instance, a memristive dot product engine had been proposed[4], which utilized analog resistive switching characteristics of the memristors and implemented the complex multiplication operation required in neuromorphic computing through Kirchhoff's Law and Ohm's law in memristive arrays. Another application of memristors leverages their intrinsic stochastic characteristics in their resistive switching, with examples including stochastic and biomimetic neural network hardware[5–7] and probabilistic computing[8–11] or true random number generators[12,13]. In this way, memristors enable a variety of emerging hardware and are discovering a variety of uses.

In addition, its utilization for security devices is noticeable[12–14]. In this application, memristors are primarily used as random number generators, leveraging their inherent stochastic switching characteristics. While this application is viable, the various functions required to operate security hardware still need to be implemented using conventional technologies. Consequently, the overall performance improvement in the entire system may not be drastic. In this context, we propose a technique for implementing homomorphic encryption hardware (HE-HW) by harnessing the diverse capabilities of memristors. Homomorphic encryption is an advanced encryption technology that allows for the application of arbitrarily complex functions to encrypted data without decryption[15,16]. This provides a solution to

[1]Department of Materials Science and Engineering, Korea Advanced Institute of Science and Technology (KAIST), Daejeon, Republic of Korea.
✉e-mail: km.kim@kaist.ac.kr

address data security concerns, particularly when dealing with data communication in cloud computing environments. After the stability of fully homomorphic encryption (FHE) was first proven by Gentry[17] in 2009, various FHE schemes such as HEAAN[18], SEAL, HElib[19], and PALISADE, etc.[20,21] have been developed. These schemes are all based on software-based mathematical cryptosystems that utilize conventional hardware. In this case, they cannot avoid the limitations posed by recent energy constraints, especially at the edges.

For the HE-HW to have advantages, it must efficiently perform two core operations required for the HE, which are (i) encryption key generation and (ii) analog vector-matrix multiplication (VMM)[17,22,23]. Implementing these characteristics individually with respective memristors is feasible[5,24–26]. However, if they are integrated into separate chips, reading the encryption and decryption keys is necessary for data transfer between the chips, exposing them to the risk of side-channel attacks[27–30]. Furthermore, their memory sizes are fixed, so their flexible utilization is not feasible. Whereas, when a single chip can perform both operations, it can directly execute VMM and store the outputs without the need for the key readout process, thereby eliminating the security risks. In addition, one can flexibly partition their sizes, making more efficient hardware utilization feasible. Nevertheless, achieving simultaneous implementation of these characteristics within a single memristor remains a challenging problem.

Here, we propose a Pt/Ta$_2$O$_5$/Mo metallic cluster-type memristor (Mo-MCM) that exhibits low-current and analog operations similar to conventional metallic cluster-type memristors[31–34] but additionally with an enhanced stochastic switching characteristic, making it suitable for the core component of the HE-HW. In Mo-MCM, we identified that the set switching process is highly stochastic due to the intrinsic high interfacial energy of Mo, which complicates and randomizes the ionization process from the active electrode[35,36]. This results in a wide switching voltage variation, enabling its use in random number generation. Although the cluster formation process is stochastic, once it is formed, its size can be accurately controlled by a compliance current, enabling analog conductance state programming. This analog memory capability allows for the storage of both encrypted and decrypted input data and facilitates more efficient execution of VMM. We confirm its digital random number generation and analog conductance programming capabilities from an integrated 1-Transistor-1-Memristor (1T1M) array device. Then, we demonstrate the working of core operations experimentally and suggest the entire process of performing homomorphic encryption using the in-memory computing of hardware. We achieved ~90% data similarity between the conductance values stored in the Mo-MCM through analog VMM computation and the ideally calculated value, indicating minimal data corruption errors. This highlights the potential for memristive HE-HW.

## Results
### Stochastic metallic cluster formation dynamics of Pt/Ta$_2$O$_5$/Mo memristors

To implement memristors in HE-HW, one must identify memristors with both stochastic switching characteristics and analog resistance states. However, a single memristor with these properties does not currently exist. Therefore, we attempted to discover such memristors based on existing ones. Among various types of existing memristors, we focused on a Pt/Ta$_2$O$_5$/Ru metallic cluster-type memristor (Ru-MCM) proposed by Yoon et al.[34], exhibiting low current analog characteristics with high CMOS compatibility. In this device, the switching mechanism is attributed to the repeated formation and dissolution of Ru nanoclusters generated from active electrodes. Unlike filament type switching (or electrochemical metallization), where continuous filament bridges form, the clusters in this device are discontinuous. As a result, electrical conduction relies on a hopping mechanism, allowing operation in a low current range. In addition, the gradual formation of clusters makes this device well-suited for analog switching. However,

the formation process of Ru nanoclusters is insufficiently stochastic to be used for the encryption key generator. We, therefore, sought to understand the nanocluster formation process and explored mobile species that inherently introduce stochasticity into this process. As a result, we identified Mo-MCM, where Mo, as a mobile species, can provide such functionalities. Figure 1a illustrates the suggested switching mechanism of the Mo-MCM.

Figure 1b shows a top-view optical microscopy image of a single Mo-MCM. We also integrated Ru-MCM for comparison. The cross-section area is $5 \times 5 \mu m^2$. A detailed device fabrication process can be found in the Experimental Section. The direct current (DC) resistive switching current-voltage (I–V) curve of Mo-MCM is shown in Fig. 1c. In the set-switching (switching from a high resistance state (HRS) to a low resistance state (LRS)), a 4 V was applied with a compliance current (I$_{cc}$) of 1 μA. In the reset-switching from the LRS to the HRS, a −3.5 V was applied without the I$_{cc}$. The low current operation below 1 μA and gradual set-switching behaviors without electroforming indicate the switching mechanism is attributed to the electrochemical metallization-based nanocluster mechanism[33,37].

During the I–V cycles, the Mo-MCM exhibited non-uniformity in the HRS resistances (R$_{HRS}$) and set voltages (V$_{SET}$) but no coherence between them. Generally, the V$_{SET}$ is highly dependent on the R$_{HRS}$; as the HRS is more resistive, the V$_{SET}$ gets higher. However, in our device, the V$_{SET}$ - R$_{HRS}$ plot showed no correlation, inferring the set-switching is intrinsically stochastic. This trend is not observed in Ru-MCM (See Supplementary Fig. S1 for the Ru-MCM's non-stochastic behaviors.) Thus, we consider that it is an intrinsic characteristic of the Mo-MCM.

So, we investigated the factors contributing to the relatively higher stochastic characteristics of Mo-MCM associated with the physical behavior of nanoclusters and their mechanisms of formation and dissolution[38]. To understand the set and reset switching dynamics in the Mo-MCM (and Ru-MCM for comparison), we adopted the field-induced nucleation theory model[39–42]. This theory provides a means to estimate the metallic cluster formation process by electrical measurement.

In theory, the nanoclusters can only achieve stability by surpassing the nucleation energy barrier, $W(E)$ resulting in stable nanocluster formation when the nucleus radius exceeds the critical radius $R_E$. The $W(E)$ can be described as follows.

$$W(E) = W_0 \alpha^{3/2} \frac{E_0}{E} = W_0 \alpha^{3/2} \frac{E_0 d}{V} \quad (1)$$

where $W_0$ is the zero-field nucleation barrier energy, $\alpha$ is a geometric factor (≈0.5), $E_0$ is a voltage acceleration factor (≈1MV/cm), $d$ is the thickness, and $V$ is the applied voltage. Since $W_0$ represents the stability of the cluster, it is directly associated with the available size of the cluster; as $W_0$ is higher, the available cluster size in the oxide can be bigger. Also, it is known that the switching delay time ($\tau_d$) shows a close correlation with the $W_0$ by the following equation[13,39,41,43]:

$$\tau_d = \tau_0 \exp\left(\frac{W(E)}{k_B T}\right) = \tau_0 \exp\left(\frac{W_0 \alpha^{3/2} E_0 d}{k_B T V}\right) \quad (2)$$

where $\tau_0$ is a vibrational time that is treated as constant.

The $\tau_d$ is the time delay of the current response, and it can be examined from a time-dependent I–V measurement. Figure 1d shows one of the results of the $\tau_d$ measurement in Mo-MCM, where the black line represents applied voltage, $V$, and the red line represents current response, $I_{out}$. The current response after a single pulse is saturated through a section that increases linearly. Here, $\tau_d$ is a time when the output current reaches the saturation value at a given applied voltage. (The $\tau_d$ measurement for Ru-MCM can be found in Supplementary Fig. S2).

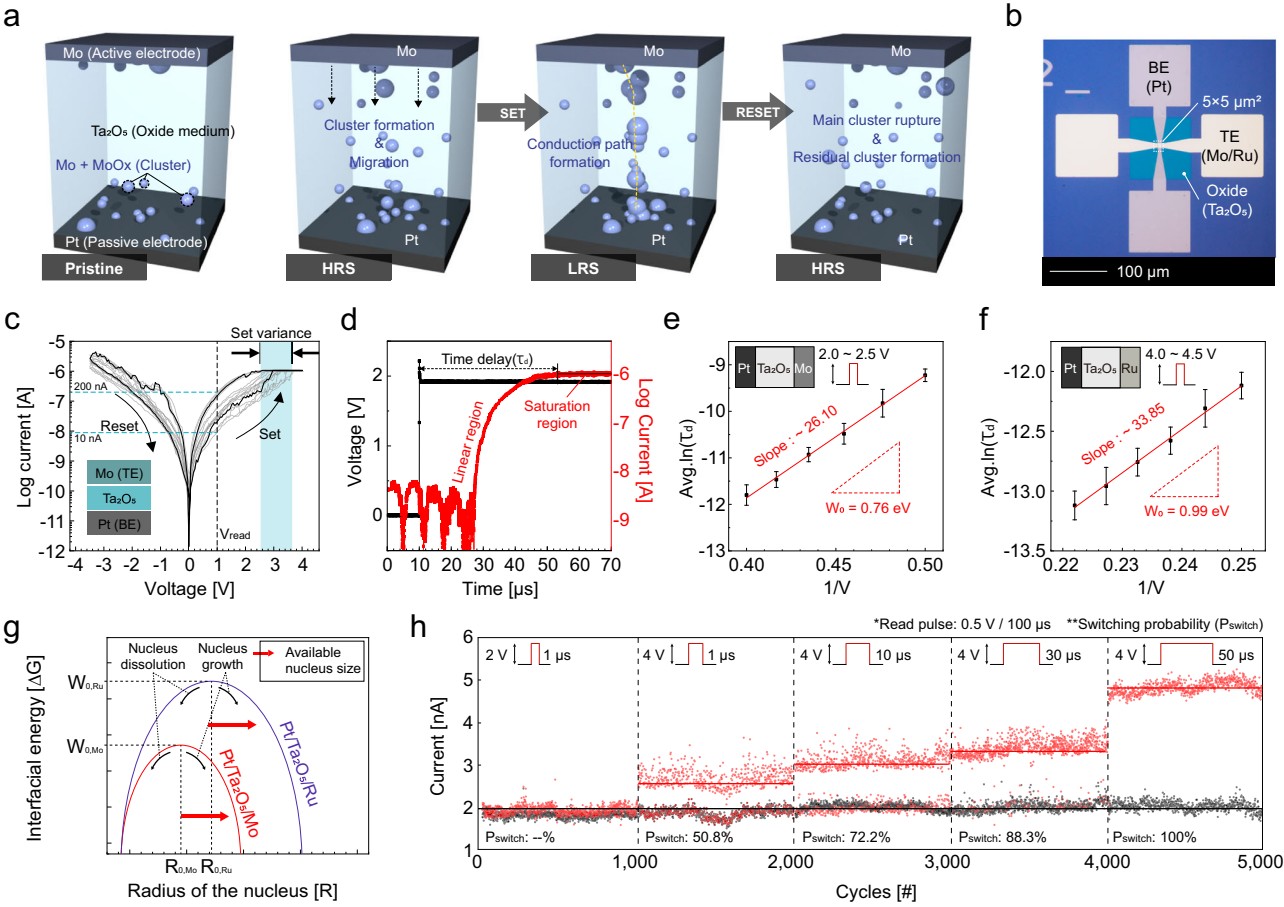

**Fig. 1 | Analysis of switching mechanism and electrical characteristics of Pt/Ta₂O₅/Mo memristor (Mo-MCM). a** Schematic images for the repeated formation and dissolution of cation nanoclusters generated from Mo active electrodes in Mo-MCM. **b** A top-view optical microscopy image of a single Mo-MCM. **c** The resistive switching I–V curve of Mo-MCM. **d** A time-dependent current-voltage measurement for Mo-MCM. **e**, **f** $\tau_d$ – 1/V plot results for Mo-MCM and Pt/Ta₂O₅/Ru

memristor (Ru-MCM) with $W_0$ obtained from the slope. **g** A graph comparing the cluster nucleus size differences based on $W_0$ in Mo-MCM and Ru-MCM. **h** Set switching probability according to the pulse conditions. As the set pulse width increases, both the switching probability ($P_{switch}$) and the programmed conductance increase.

After obtaining $\tau_d$ at varying applied voltages and plotting $\tau_d$ vs 1/V, $W_0$ can be obtained from the slope of the plot by Eq. (3). Figure 1e, f are $\tau_d$ - 1/V plot results for Mo-MCM and Ru-MCM, respectively. Each data point on the plot was obtained from 30 devices (a total of 180 cycles for a single device) with a measurement range of 2.0 - 2.5 V for Mo-MCM and 4.0 ~ 4.5 V for Ru-MCM. By linear fitting, the $W_0$ of the Mo-MCM was identified as 0.71 eV, while that of the Ru-MCM was 0.97 eV. The different magnitudes of $W_0$ infers that the size of available clusters in Mo-MCM has a broader range and starts from a smaller size, as depicted in Fig. 1g. Consequently, it can be inferred that the residual clusters in Mo-MCM may exhibit a distribution ranging from smaller to larger sizes. This results in a more diverse set of conducting paths, in contrast to Ru-MCM, resulting in greater stochastic switching characteristics.

The distinctive feature of Mo-MCM is that its set switching is stochastic, but once it is set-switched, it shows stable analog states. Figure 1h shows the transition behavior from stochastic switching to deterministic switching. Here, the set-switching pulse widths were changed from 1 μs to 50 μs at 4 V. At a weak set condition, the device stochastically set-switched, so it randomly remained in the HRS even after receiving the set-switching pulse. The switching probability was increased as the pulse width increased, and eventually, a stable set- and reset-switching was obtained. Also, the conductance of the LRS gradually increased even during stochastic switching, suggesting the analog state programming capability of Mo-MCM. (The potentiation characteristics under 1 μs to 50 μs at 4 V are shown in Supplementary Fig. S3.)

The deterministic switching by 50 μs at 4 V corresponds to the lowest conductance state, and higher conductance states can be obtained, which is demonstrated in detail later. In addition, the endurance of $10^6$ cycles was confirmed, making it sufficiently usable for HE-HW applications. (Endurance data can be found in Supplementary Fig. S4.)

The stochastic set-switching characteristic can be used for the random number generator by defining the LRS (1) and HRS (0) states for digital output. To ensure its randomness, we conducted the NIST randomness suite tests using 1000 bits – 100 sequences condition[12,13,44]. At the optimized set condition (4 V, 1 μs), the output data string passed all of the 14 NIST test criteria (see the detailed results in Supplementary Table S1). This test was performed at highly controlled conditions to demonstrate a Pseudo Random Number Generator (PRNG) capability so that our device can certainly be used as a reliable random number generator, which is sufficient for utilization in homomorphic encryption.

## Mo-MCM 1T1M array integration and evaluation

Next, we integrated a 16 × 24 1T1M array to evaluate the HE-HW feasibility. Figure 2a shows a photograph of the integrated die on a glass substrate (left panel), whose size is 3 × 3 cm², and the optical microscopy image of the core area (right panel), where each cell has a size of 70 × 65 μm², including a Mo-MCM of 5 × 5 μm². To account for the low-current features of Mo-MCM, a thin film transistor process using the amorphous silicon (a-Si) as a channel layer was adopted from a

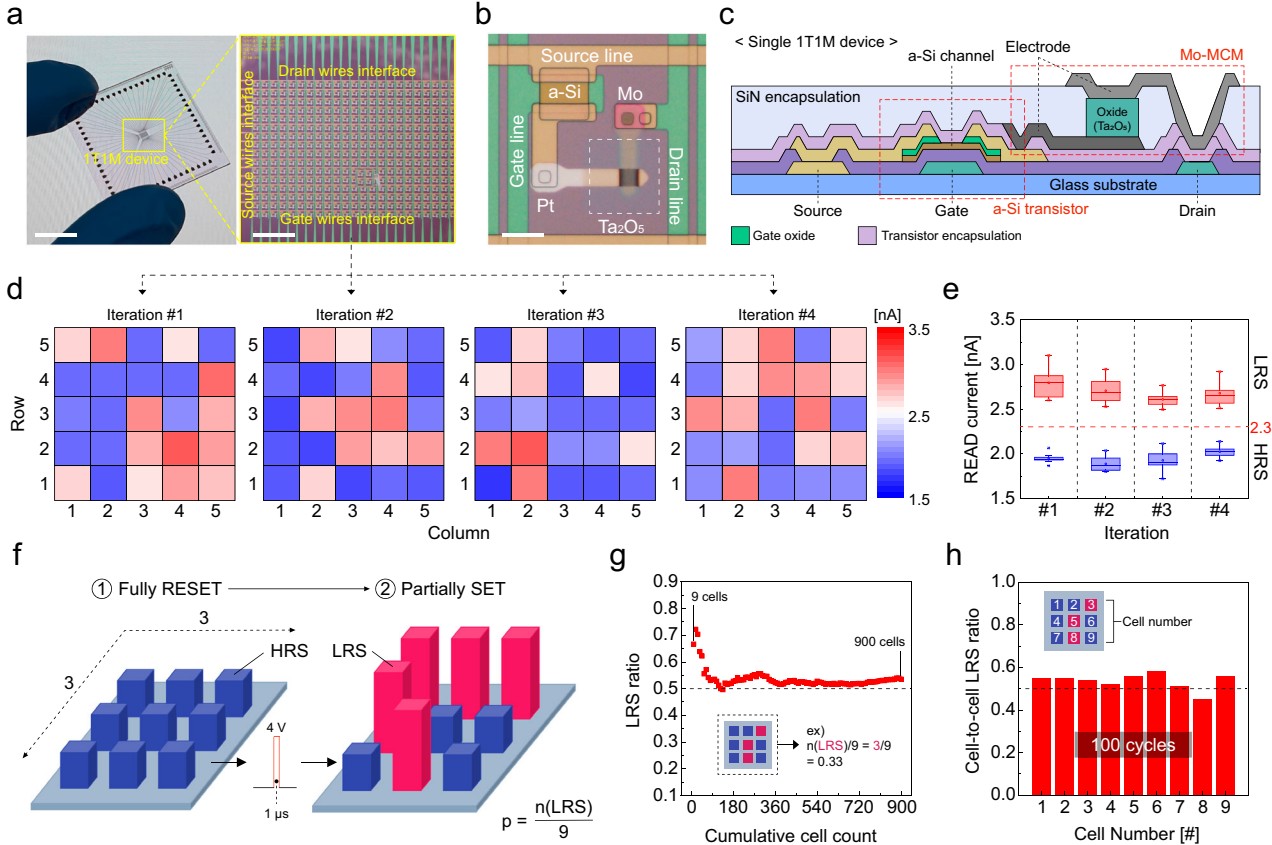

**Fig. 2 | 1T1M array integration of Mo-MCM and its evaluation on uniformity. a** A photograph of the 1T1M device on a glass substrate and an optical microscope image of the cells at the core area. Scale bars indicate 1 cm (left) and 250 μm (right). **b** A magnified optical microscope image of a single 1T1M cell. A scale bar indicates 10 μm. **c** A schematic image of the cross-section of the 1T1M device. **d** Heatmaps for 4 iterations of random number generation from stochastic set-switching in 5 × 5 sub-arrays. **e** Current distribution of LRS and HRS on each iteration. **f** Illustration of the process for calculating LRS ratio by stochastic set-switching. **g** A graph for the ratio of LRS after stochastic set-switching as a function of the cycling number. **h** A graph for the stochastic set-switching uniformity of 9 cells across 100 cycles.

commercial foundry. More data on the transistor performance can be found in Supplementary Fig. S5. Figure 2b shows the cell image, and Fig. 2c illustrates the cell configuration.

From the array, we have demonstrated the random number generation. Figure 2d shows four samples of random number generation from a 5 × 5 sub-array by applying 4 V, 1 μs of $V_{SET}$. After set-switching, the devices were stochastically set-switched, and the LRS and HRS could be clearly distinguished. Their distribution is statistically summarized in Fig. 2e. The results confirmed that array-level random number generation was successfully achieved.

To verify cycle-to-cycle and cell-to-cell random number generation uniformity, stochastic set-switching was repeatedly monitored on a 3 × 3 sub-array for 100 times (Fig. 2f). Figure 2g shows the ratio of LRS after the stochastic set-switching as a function of the cycling number. As the cycles repeat, the LRS ratio approaches the standard of pseudo-true random numbers, which is a 50% LRS ratio. Figure 2h illustrates the stochastic set-switching uniformity of 9 cells across 100 cycles. The cell-to-cell uniformity of each cell position showed an LRS ratio close to 50% over the 100 cycles in a distinguishable state (More data on the device-to-device uniformity can be found in Supplementary Fig. S6). These results indicate that the random number generation of Mo-MCM is uniform and reliable.

### Analog conductance states and VMM computations characteristics of 1T1M device

Next, we demonstrated array-level analog resistive switching behavior. Figure 3a shows a photograph depicting the testing environment, which allows access to individual cells for programming multiple analog

conductance states in the 1T1M devices and also multiple access for executing VMM computations. More details on the testing platform, including both hardware and software interfaces, are depicted in Supplementary Fig. S7. Figure 3b, c show a conductance map for 3-bit analog states implemented in a 5 × 5 array and their uniformity, respectively.

Figure 3d represents the multiply-and-accumulate (MAC) circuit diagram for VMM computations. With a constant pulse length of 0.5 ms, different input voltage vectors $V_1$ and $V_2$ were applied to 2 cells with conductance levels $G_1$ and $G_2$ during the inferencing, resulting in the summation of current values ($I_{out}$) performed through Ohm's law and Kirchhoff's current law, according to Eq. (3).

$$I_{out,j} = \sum_{i=1}^{n} V_{in,i} G_{ij} \tag{3}$$

During this process, 10 V of gate voltage ($V_G$) was applied to the transistors of 2 cells in a 1T1M device to enable ON switching.

Figure 3e, f represent colormaps of the output current corresponding to $V_1$ and $V_2$ for $G_1$ and $G_2$. In Fig. 3e, $G_1$ was programmed to 25 nS, which is HRS, followed by $G_2$ with 5.8 nS and 12 nS. Similarly, in Fig. 3f, $G_1$ was programmed to 143 nS, which is LRS, followed by $G_2$ with 25 nS and 58 nS. As a result, the VMM computations across the entire range were depicted, demonstrating that 1T1M Mo-MCM can perform VMM computations reliably in the range between HRS and LRS.

### Homomorphic encryption system configuration

By using HE-HW, various functionalities required for the homomorphic encryption process can be applied through the following data

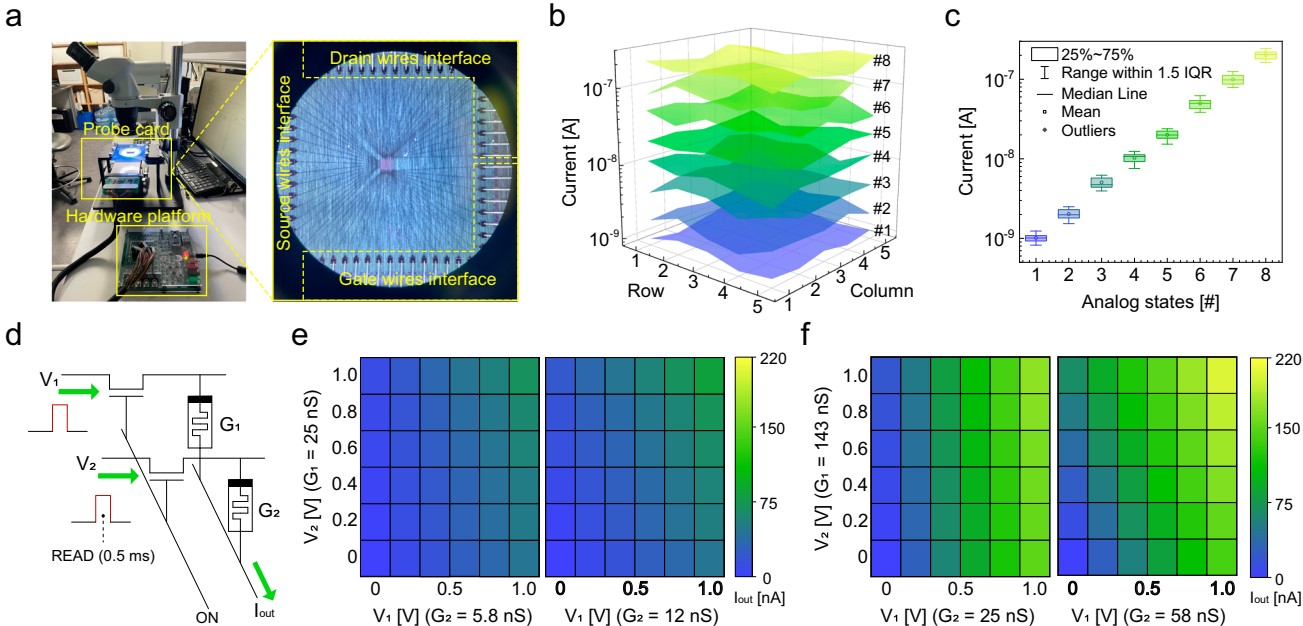

**Fig. 3 | Analog conductance states and VMM computations characteristics of 1T1M device. a** Photographs for testing environment, which allows access to individual cells for programming multiple analog states. **b** A conductance map for a 3-bit analog state implemented in a 5 × 5 sub-array. **c** A graph for indicating cell-to-cell uniformity of each analog state. **d** Multiply-and-accumulate (MAC) circuit diagram for VMM computations. **e**, **f** Colormaps of the output current corresponding to $V_1$ and $V_2$ for $G_1$ and $G_2$.

communication procedure on a single device. Figure 4a, b illustrate the requesting step, where the client requests a task from the cloud, and the responding step, where the cloud performs the requested task and returns the results to the client, respectively. In these processes, the HE-HW plays a role in both encrypting and decrypting data. Here, it needs to satisfy the following conditions to be used as a hardware accelerator: First, the HE-HW must be capable of generating random numbers to create encryption keys internally. These encryption keys should be kept confidential and must not be accessible from external sources. Second, the HE-HW should be able to encrypt the raw data and decrypt the processed encrypted data. For this, the HE-HW should be able to store the encrypted data and the processed encrypted data. This stored data is used to communicate with external entities such as the cloud and clients. Figure 4c summarizes practical scenario-based explanations: the client first encrypts (*Enc*) their data through HE-HW (1) and then forwards the encrypted data to the host (2) as depicted in Fig. 4a. Each of these processes (1-2) corresponds to steps (i-ii) in Fig. 4c. Subsequently, the host executes computations (*f*) on the encrypted data (3), which corresponds to steps (iii) in Fig. 4c. The client then receives the data (4) and decrypts ($Enc^{-1}$) using HE-HW (5). In this case, each of these processes (3−5) corresponds to steps (iii−v) in Fig. 4c. Even after decryption is performed on the client side, the computation *f* performed on the raw data *m* can still be observed (*f(m)*), making it possible to maintain data privacy and security during processing. Also, it is worth noting that the host conducting computations only deals with encrypted data and cannot access the raw data.

Figure 4d shows the role of HE-HW in this process. Before clients input the raw data into HE-HW, the raw data undergo one-dimensional encoding suitable for the subsequent VMM computations. At the same time, HE-HW generates an encryption key through stochastic set-switching, followed by generating the decryption key via a peripheral circuit. Consequently, both encryption and coupled decryption keys are stored in the array, but they should not be accessible from outside. Then, it executes the VMM computations between the input data and the encryption key and stores the results inside the array, which is accessible. This encrypted data is then sent to the host. Then, the host processes it and returns the processed data to the client. Then, the client decrypts it through the VMM computations using the coupled decryption key.

## Demonstration of homomorphic encryption system on 1T1M device

Next, we demonstrated the full operation of the homomorphic encryption using the 1T1M HE-HW, as shown in Fig. 5a. Here, we performed an addition operation on 3 × 3 matrices using binary input. The 3 × 3 matrices are denoted as *x* and *y* represent the raw data from the client side, as well as *z* represents the expected results for addition operation. The encryption keys, Key A and Key B, generated from the 1T1M device by pulses for stochastic switching (4 V, 1 μs), are stored internally as binary matrices. Each encryption key can transform the raw data *x* and *y* then encrypt into 4 sets (*x*\*Key A, *x*\*Key B, *y*\*Key A, *y*\*Key B) by VMM computations. The multiplication operation results of *x* and *y* are represented by blue and purple lines, respectively. The encryption keys are also applied to the expected addition operation result *z*. The encryption results for this (*z*\*Key A, *z*\*Key B) are compared to the addition of the encrypted results for *x* and *y* ((*x* + *y*)\*Key A, (*x* + *y*)\*Key B), matching the expected operation, indicated by the green dashed line. Data consistency comparisons are presented in Fig. 5b. The average preservation rate of addition in the data for Key A and Key B is denoted as 91.2% and 94.7%, respectively. Here, errors arise from variations in the analog states, particularly those in the higher conductance states, which significantly impact the VMM outputs. To enhance the preservation rate, achieving more uniform analog states is crucial. This can be accomplished through the use of a more advanced CMOS process or by implementing sophisticated programming schemes[45,46].

The methodology for calculating the inverse matrix of binary matrices was previously demonstrated by Sun, Zhong et al.[47], using a small-scale peripheral circuit. The detailed circuit diagram is depicted in Supplementary Fig. S8. We adapted this method to generate the inverse matrices for encryption keys A and B through LT-Spice calculations. As the encryption process in this demonstration was simplified

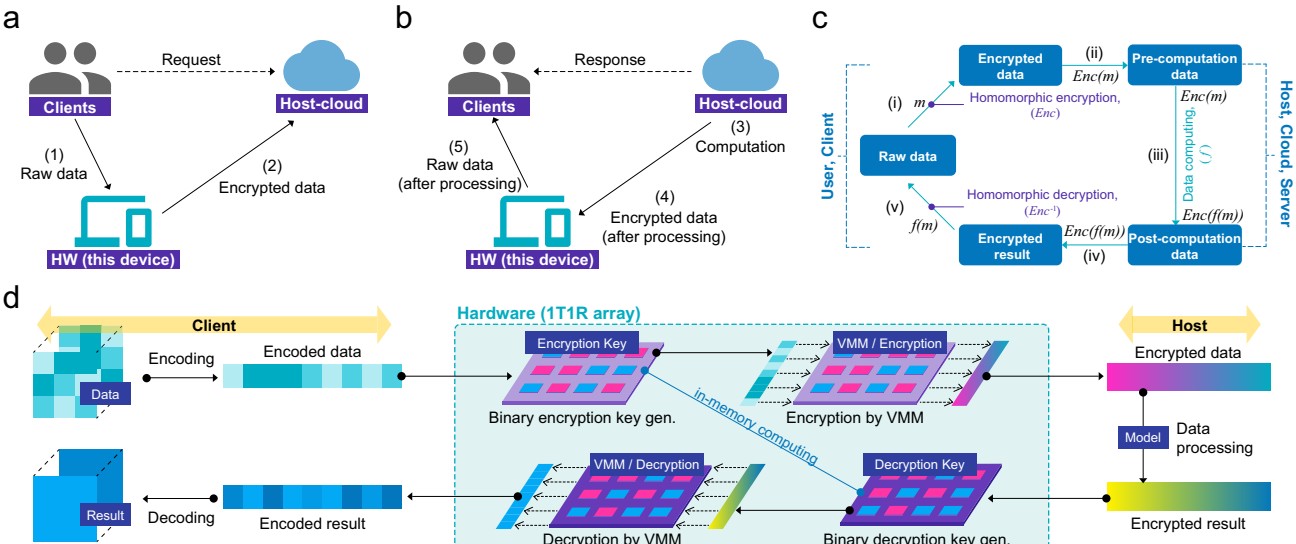

**Fig. 4 | Schematic images of various homomorphic encryption functionalities achievable with HE-HW and the related data communication process.**
**a**, **b** Illustrations of the sequences where the client requests a task from the cloud and the sequences where the cloud performs the requested task and returns the results to the client. **c** A diagram summarizing functionality based on mathematical expressions. **d** An illustration of the roles of HE-HW in performing homomorphic encryption functionality.

by employing a single VMM computation, the generated inverse matrices can serve as decryption keys. Therefore, they were stored alongside the encryption key within the 1T1M array. To verify the preservation of addition in the encrypted data, decryption was conducted through VMM computations after the addition operation $((x+y)*\text{Key A}*\text{Key A}^{-1}, (x+y)*\text{Key B}*\text{Key B}^{-1})$, and the results were compared to the addition operation result, $z$, from the raw data (Fig. 5c). The average preservation rate of encryption followed by the decryption process in the data for Key A and Key B is denoted as 89.2% and 91.8%, respectively.

In such a process, maximizing the use of the available storage window is the best way to achieve efficient storage of encrypted data. While data after encryption, such as $y*\text{Key B}$ in Fig. 5a, may be discretely separated, the resulting values may not align with the most efficient memory window of the Mo-MCM, as shown in Fig. 4c. During the demonstration, we discretely quantized the data that went through the encryption process and stored it as 3-bit data, as depicted in Fig. 5d. To perform the process as described, a controller is required that is capable of reordering data within the memory window before and after matrix operations. This controller would play a similar role to a NAND controller.

Figure 5e illustrates the memory architecture of the HE-HW accelerator using Mo-MCM. Homomorphic encryption technology requires the memory architecture to be divided into two distinct areas within the crossbar array to effectively carry out its two crucial functions: key generation for encryption and decryption and storage of encrypted data. The memory area responsible for key generation for encryption and decryption must remain isolated, with no external exposure, and only internal processor-initiated data transfers should be allowed. On the other hand, a dedicated cache area should be allocated for the storage of encrypted data and results for operation, which allows for I/O access to the data after the encryption process. By distinguishing between accessible and inaccessible areas, as depicted in the diagram, an ideal configuration for the HE-HW accelerator using Mo-MCM can be achieved.

## Discussion

In this paper, we proposed a 1T1M array embodying a Pt/Ta$_2$O$_5$/Mo-structured Mo-MCM as a HE-HW accelerator. The Mo-MCM, which

operates as a low-current, cluster-type component, unlike the filamentary type, could perform both random number generation and VMM computations by manipulating input pulses. We also analyzed the physical behavior of the metallic cluster-type memristors by comparing mobility and interfacial energy and concluded that the Mo cluster formation process could be intrinsically stochastic. Then, we integrated it into the 1T1M array and proved its use as a HE-HW accelerator through full experimental demonstration.

Prior research has predominantly focused on integrating memristors with varying characteristics to develop sophisticated hardware systems. Our study sets itself apart by proposing a method to exploit the multiple inherent properties of a single memristor in complex hardware setups. We believe this research will establish a foundation for the wider adoption of memristors, which display a range of diverse and compelling phenomena in various emerging hardware fields.

Nevertheless, the proposed devices and systems have the following potential improvements. Mo-MCM's switching speed ranges from $1\,\mu s$ for stochastic switching to $50\,\mu s$ for deterministic switching, which is consistent with typical speeds reported in other literature. Therefore, improving the switching speed appears to be a challenging task. While this speed may be slow for dynamic memory applications, it does not significantly hinder its functionality in HE-HW, where stochastic switching and analog programming are advantageous.

Next, we demonstrated additive HE with a preservation rate of about 90%, which needs to verify whether it can be said to be Fully HE through multiplicative HE demonstration. In order to demonstrate multiplicatively HE, which requires complex functionality and precise operations, the device's analog state uniformity must be improved than now. These strategies will require further in-depth research to fully realize their potential.

## Methods

### Fabrication of 1T1M Pt/Ta$_2$O$_5$/Mo device

The 1T1M arrays are composed of Pt/Ta$_2$O$_5$/Mo memristors with a-Si channel transistors. After fabricating the transistor array of a-Si channel using a commercial foundry, memristors were used as a post-process. A Pt (40 nm)/Ta$_2$O$_5$ (20 nm)/Mo (40 nm) structure was fabricated on a SiO$_2$/Si substrate by the following procedure. The bottom Pt

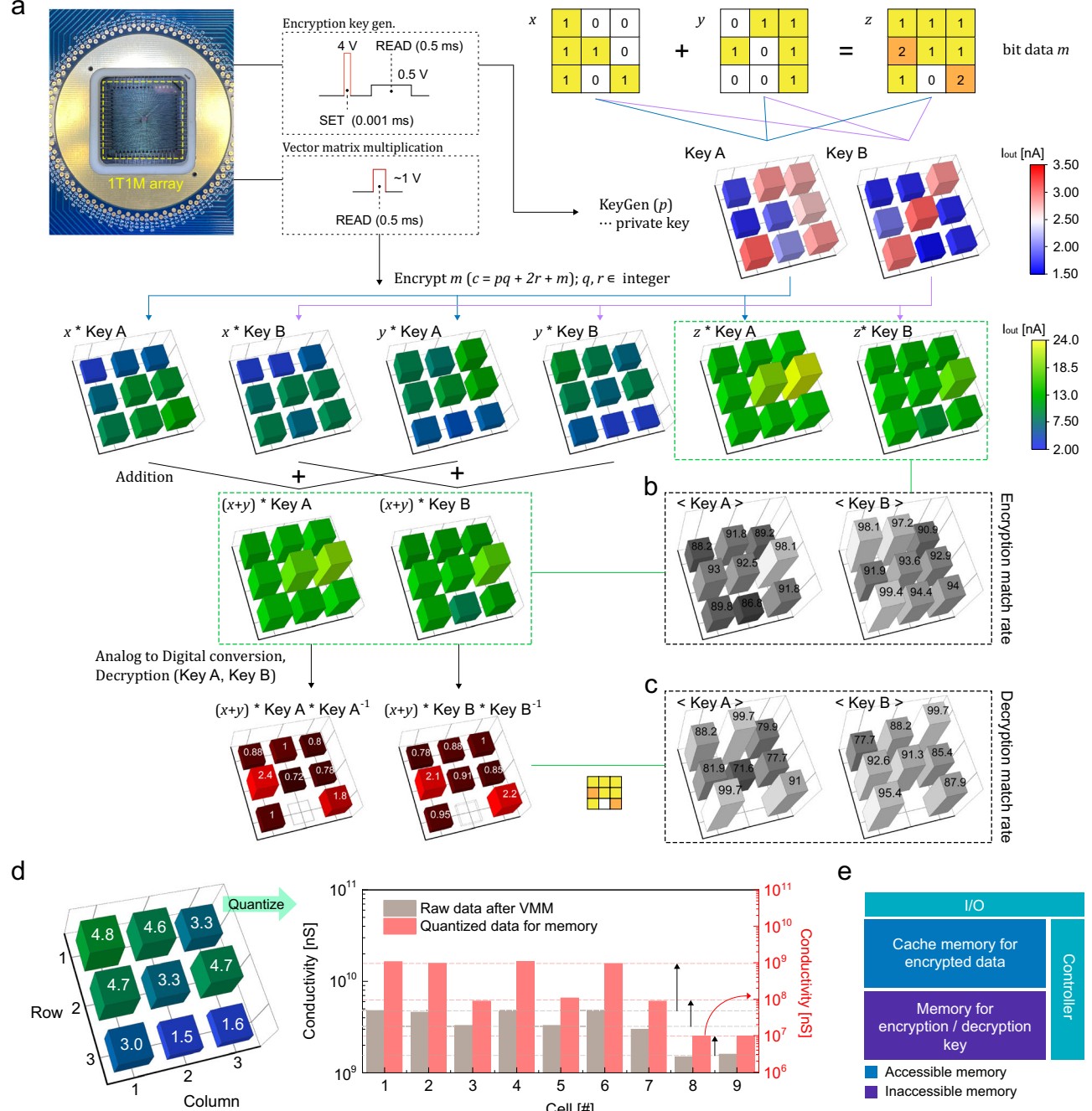

**Fig. 5 | Demonstration of various capabilities in homomorphic encryption. a** An illustration of a demonstration showcasing homomorphic encryption via HE-HW for addition operation on 3 × 3 matrices using binary input. **b**, **c** Comparison of data consistency of encryption match rate (**b**) and decryption match rate (**c**) between simulated calculations and actual measurements. **d** Sequence for storing quantized data as 3-bit data that went through the encryption process. **e** An illustration of the hardware architecture of the HE-HW accelerator.

electrode was deposited by E-beam evaporation with an adhesion layer of Ti (5 nm). The $Ta_2O_5$ oxide layer, which acts as a solid electrolyte, was deposited by a DC reactive sputtering using a Ta metal target with oxygen gas. The sputtering power was 150 W at 150 °C. The top Mo electrode was deposited by E-beam evaporation. Each layer was patterned by lift-off processes to form $5 \times 5\ \mu m^2$ cross-junctions.

### Electrical characterization
The electrical characterization was performed using Keithley 4200A-SCS, and a hot chuck controller (MST-1000H) was used for the ambient temperature control. During the electrical measurement, the top electrode was biased while the bottom electrode was grounded. The

electrical characterization of 1T1M devices was performed using custom-made measuring equipment based on ArC ONE from ArC instrument©.

## Data availability
All data generated in this study are provided in the article and its Supplementary Information. Additional data are available from the corresponding author upon request.

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

## Acknowledgements

This research was supported by the National Research Foundation of Korea (NRF) (Grant numbers: RS-2023-00216619, RS-2023-00216992, 2022M3F3A2A01076569, 2022M3I7A4085484, and 2023R1A2C2005159), NNFC (Grant number: 1711160154), and UP program of KAIST (Grant number: N10230061).

## Author contributions

The project was conceived and designed by W.H.C. NIST test for device randomness was performed by J.H.I. Device fabrication and evaluation were performed by J.B.J., G.K. Testing of electrical characteristics of individual components was performed by W.H.C. and J.B.J. Testing of electrical characteristics of array devices was performed by W.H.C. The manuscript has been written by W.H.C. and K.M.K. A detailed review of the manuscript was done by K.M.K.

## Competing interests

The authors declare no competing interests.
