## [Peer Review File · Nature Communications]

REVIEWER COMMENTS

Reviewer #1 (Remarks to the Author):

This manuscript introduces a Pt/Ta₂O₅/Mo metallic cluster-type memristor (Mo-MCM) that can be used for random number generation and analog vector-matrix multiplication. Based on the Mo-MCM 1T1M prototype, the authors demonstrate an in-memory homomorphic encryption system, which provides a promising solution for homomorphic encryption on edge devices. The device characteristics demonstrated in this manuscript are very interesting, and this manuscript fully explores and utilizes these characteristics considering the application requirements of homomorphic encryption.

However, this manuscript still has some problems that need to be solved. First of all, this manuscript claims it realizes the homomorphic encryption hardware (HE-HW). But this work only focuses on the encryption of input data and decryption of results using simple MVM, SOTA homomorphic algorithms use polynomial encoding, modular computation, polynomial computations, and other non-MVM operations for data encryption/decryption. Besides, memristor-based MVM, data encryption, and PUF have been studied for many years. It would be better if the authors could make more comparisons with related work and emphasize the innovations of this paper.

Below are some detailed comments:

1. Weak motivation. One device with combined stochastic switching and analog computing features could be interesting, but practically not very necessary. There is no strong motivation for having two such features based on one device structure. In other words, there could be better stochastic switching devices or better analog computing devices.
2. The authors mention that both stochastic switching characteristics and analog resistance states are important for HE-HW. On the one hand, how to quantitatively/statistically describe or define the stochastic switching characteristics? For example, it seems that Fig. S1. (a) also shows the stochastic switching characteristics during RESET. Why this RU-MCM does not have the stochastic behaviors? On the other hand, why are analog resistance states important for HE-HW? Please explain these points in more detail in the manuscript.
3. The distinctive feature of Mo-MCM is that its set switching is stochastic, but once it is set-switched, it shows stable analog states. If switching is stochastic, deterministic switching, e.g., weight update, would be costly to achieve. It could be very impractical for analog in-memory computing.

4. In Fig. 1e and Fig. 1f, the authors obtain the data from only ten devices. I am wondering whether ten devices are enough to measure a sufficiently accurate $\tau d^{-1}/V$ curve, although ten devices can also reflect the differences in trends to some extent.
5. According to Fig. 1i, the R ratio of the proposed memristor is only 2~3. Could this low R ratio result in the accuracy loss? Furthermore, the average preservation rate is about 90%. Please explain the sources of the error and how to solve this problem.
6. Relatively low performance in terms of switching speed under a practical voltage range (several microseconds under a few volts).
7. Lack of the very important endurance characteristics with only about 1000 switching operations. This is critical for both random number generation and weight update.
8. The encryption and decryption key cannot be accessible from outside to ensure security. In the proposed hardware implementation, how to ensure that these keys are not accessible from the outside?
9. Typo: In lines 143 and 144, Fig. 1g should be Fig. 1i; P5: a mean to estimate  a means to estimate; P7: a LRS ratio  an LRS ratio.

Reviewer #2 (Remarks to the Author):

Authors present Mo-electrode based stochastic-switching from Ta2O5-Pt memristor device, array, and relevant application. This is a good comprehensive study, which is worth publishing with a minor revision as suggested below.

1. Details on the dynamic information (pulse width and voltage interval) of I-V hysteresis curve measurement in Fig.1c is recommended to include.
2. Reporting the read current (or conductance) vs. # of pulse such as Fig.10 and Fig.11 of [Marinella et al. IEEE Journal on Emerging and Selected Topics in Circuits and Systems 8.1 (2018): 86-101] is recommended as a supporting information for readers in the compact model area. This will also reveal whether the stochastic-switching is attributed to mainly the physical mechanism hypothesized by Fig.1g or also in part by significant nonlinearity in switching.
3. Authors are also recommended to provide Ru-memristor's raw data equivalent to Fig. 1h and the read current vs. # of pulse plot suggested in Comment#1 as a supporting information. These raw data will be more convincing to readers in addition to the hypothesis in Fig.1g.

4. In Fig.3c, 8th state shows 200 nA, which even exceeds the SET state current (w/ 0.5V read voltage) in Fig.1c (30 nA). It will be informative to mention the endurance with a simple statement (e.g., 500 cycles, 100 cycles, etc.) for readers to know how much it deteriorates compared to limited range endurance test (>1000 cycles in Fig.1i).

5. The difference between 5x5 μm^2 memristor (single), 10x10 μm^2 memristors (in 16x24 array)? According to Fig.2 and Fig.3, it seems that the memristor's conductance doesn't change much even with larger device cross section. If the underlying physics is bulk-switching, instead of filamentary-switching, an explanation on this would be helpful.

Reviewer #3 (Remarks to the Author):

Cheong et al. present a molybdenum-based cluster-type memristor to demonstrate HE. The main idea of this work is that they have developed a memristor capable of stochastic (for random key generation) and analog switching (for computing) with different programming conditions. With this dual capability, the authors experimentally demonstrated additively HE using 1T1M arrays.

The paper is well written and will interest the readers of the journal. The demonstration of HE-HW is novel. I recommend publication after considering the following points:

1. The authors claimed that “the metallic cluster formation process is intrinsically stochastic due to the slow migration characteristics of the Mo atoms on redox reactions, resulting in a wide switching voltage variation”. This can mislead the readers, as memristors based on Ag and Cu are also known to exhibit significant variation.

2. What makes the device to switch in a cluster-type rather than a filamentary-type? The authors cited ref [27] for the concept of Ru-MCM. In Ref [27], the switching mechanisms were modulated by controlling the I_{cc} . Can the authors show similar behaviors in both Ru-MCM and Mo-MCM? Please explain more about the cluster-type switching.

3. Fig. 1a does not show the switching based on Ru nanoclusters.

4. The authors should make another plot for V_{set} - V_{HRS} to show there is no correlation.

5. Please provide more information on the NIST test in the main text or supplementary. How many devices and bits were used?

6. Can the authors provide more discussion on HE-HW? For example, would performing multiple additively HE decrease the preservation rate? If so, what would limit the performance of HE? Is it also possible to perform multiplicatively HE?

[Response to Reviewer #1]

We thank the reviewer for a thoughtful review and comments. We hope that our revision responds well to the comments.

Comment #1: “Weak motivation. One device with combined stochastic switching and analog computing features could be interesting, but practically not very necessary. There is no strong motivation for having two such features based on one device structure. In other words, there could be better stochastic switching devices or better analog computing devices.”

We sincerely appreciate your thoughtful comment. Memristors inherently possess stochastic properties, which are extensively investigated for their potential application in security hardware. These studies are valued by the readers more for their potential than for immediate practical use. We believe that proposing innovative ideas with potential and demonstrating feasibility is crucial for impactful research. To emphasize this point, we have revised the introduction as follows.

Revised Manuscript (page 3):

“Another application of memristors leverages their intrinsic stochastic characteristics in their resistive switching, with examples including stochastic and biomimetic neural network hardware⁵⁻⁷, and probabilistic computing⁸⁻¹¹ or true random number generators^{12,13}. As such, memristors enable various emerging hardware, and expanding their horizons to discover new uses can be very intriguing.

Additionally, its utilization for security devices is noticeable¹²⁻¹⁴. In this application, memristors are primarily used as random number generators, leveraging their stochastic switching properties. While this application is viable, the various functions required to operate security hardware still need to be implemented using conventional technologies. Consequently, the overall performance improvement in the hardware may not be drastic. In this context, we propose a technique for implementing homomorphic encryption hardware (HE-HW) by harnessing the diverse capabilities of memristors.”

Additionally, in this study, we demonstrate the implementation of two features (stochastic switching and analog state programming) in a single device, which offers the following benefits:

1. Enhanced security: By performing encryption key generation and data encryption through in-memory computing within a single device, homomorphic encryption can be conducted without the need for a separate processor. This eliminates the encryption key readout process, thereby reducing the risk of side-channel attacks that pose security threats to encryption systems.
2. Customizability: Our device can function both for key generation and as analog memory, allowing for flexible allocation of these two components within the hardware. For instance, the hardware can be utilized entirely as a random number generator or as analog memory. This flexibility enables effective tailoring of the device to suit specific tasks.

We have added the following sentence to the introduction part to highlight these advantages.

Revised Manuscript (page 3-4):

“However, if they are integrated into separate chips, reading the encryption and decryption keys is necessary for data transfer between the chips, exposing them to the risk of side-channel attacks.²⁷⁻³⁰ Furthermore, their memory sizes are fixed, so their flexible utilization is not feasible. Whereas, when a

single chip can perform both operations, it can directly execute VMM and store the outputs without the need for the key readout process, thereby eliminating the security risks. Additionally, one can flexibly partition their sizes, making more efficient hardware utilization feasible. Nevertheless, achieving simultaneous implementation of these characteristics within a single memristor remains a challenging problem.”

Above all, our study proposes a method for utilizing multifunctionality of memristive devices. Typically, previous studies on memristors have focused on exploiting just one functionality per memristive device. This study serves as an example of the greater potential of memristor devices and will contribute to enhancing their applicability.

We added the following sentence in the discussion part as a remark on the significance of our study.

Revised Manuscript (page 11):

“Lastly, we would like to highlight the significance of this study and identify potential challenges that need to be addressed. Prior research has predominantly focused on integrating memristors with varying characteristics to develop sophisticated hardware systems. Our study sets itself apart by proposing a method to exploit the multiple inherent properties of a single memristor in complex hardware setups. We believe this research will establish a foundation for the wider adoption of memristors, which display a range of diverse and compelling phenomena, in various emerging hardware fields.”

Comment #2: “The authors mention that both stochastic switching characteristics and analog resistance states are important for HE-HW. On the one hand, how to quantitatively/statistically describe or define the stochastic switching characteristics? For example, it seems that Fig. S1. (a) also shows the stochastic switching characteristics during RESET. Why this RU-MCM does not have the stochastic behaviors? On the other hand, why are analog resistance states important for HE-HW? Please explain these points in more detail in the manuscript.”

In this study, we focused on stochastic SET switching specifically. Initially, the pristine device state is in the high resistance state, making it natural to generate random numbers through set switching. This approach is consistent with the methods applied in most previous studies¹⁻³. In this regard, we believe that utilizing reset switching is inefficient and thus it is not of interest to this study. Moreover, set switching involves clustering processes that are dependent on the material’s interfacial energy between active metal and metal oxide layer, leading to significant variations depending on the material properties. As shown in the figures below, Mo-MCM exhibited stochastic switching characteristics with very small conductance changes (left panel), whereas Ru-MCM demonstrated deterministic switching characteristics during similar conductance changes (right panel).

To quantitatively compare the degree of stochastic set switching, we believe that the NIST randomness test suite results are optimal. Our proposed Mo-MCM passed 14 out of 15 tests, thus qualifying it as a random number generator (Table S1). Whereas, in Ru-MCM, stochastic switching is almost undetectable, therefore it cannot pass the NIST test. Based on these findings, we conclude that Mo-MCM has the capability of generating encryption keys, while Ru-MCM does not.

To address these points, we have made additional revisions to the manuscript and supplementary information.

In the revised manuscript, we elaborated with the NIST test results as follows.

Revised Manuscript (page 7):

“The stochastic set-switching characteristic can be used for the random number generator by defining the LRS (1) and HRS (0) states for digital output. **To ensure its randomness, we conducted the NIST randomness suite tests using 1,000 bits – 100 sequences condition^{12,13,44}.** At the optimized set condition (4V, 1 μs), the output data string passed all of the 14 NIST test criteria **(see the detailed results in Supplementary Table S1)**. This test was performed at highly controlled conditions to demonstrate a **Pseudo Random Number Generator (PRNG) capability so that our device can certainly be used as a reliable random number generator, which is sufficient for utilization in homomorphic encryption.**”

Also, we have added two supplementary information.

In Table S1, we added NIST randomness test results of Mo-MCM as follows.

Revised Supplementary Information (page 2):

Table S1. NIST **randomness suit** test results for evaluating random number generation performance required for homomorphic encryption. **Each test for Mo-MCM was conducted under the condition of 1,000 bits - 100 sequences.**

Test #	Test name	P-value (Mo-MCM)	Proportion (100 seq.)	Result (Mo-MCM)
1	Frequency (Monobit)	0.011412036	0.98	Pass
2	Frequency within a block	0.550717949	0.99	Pass
3	Runs	0.532759772	1.00	Pass
4	Longest run of ones in a block	0.348025369	0.98	Pass
5	Binary matrix rank	0.015710521	0.98	Pass
6	Discrete Fourier transform (Spectral)	0.771670504	0.99	Pass
7	Non-overlapping template matching	1.00	1.00	Pass
8	Maurer's universal	0.988785497	1.00	Pass
9	Linear complexity	0.999995031	1.00	Pass
10	Serial	0.761854261	0.99	Pass
11	Approximate entropy	0.999520079	0.99	Pass
12	Cumulative sums	0.017345555	0.97	Pass
13	Random excursions	0.950167133	0.98	Pass
14	Random excursions variant	0.910979293	0.99	Pass

Also, we added Ru-MCM characteristics and compared Mo-MCM and Ru-MCM switching characteristics as follows.

Revised Supplementary Information (page 3):

Fig. S1. Electrical characteristics of Ru-ECM (Pt/Ta₂O₅/Ru) memristors. a) I-V characteristic of a Ru-ECM memristor with 5 repetitions, which shows stable and repetitive switching behavior. b) Schematic image for a vertical structure of Ru-ECM memristors. The bottom Pt electrode was deposited using E-beam evaporation with a thickness of 40 nm, followed by a deposition of 20 nm of Ta₂O₅ using DC reactive sputtering with a Ta target. The top Ru electrode was deposited with a thickness of 40 nm using DC sputtering. c-d) Comparison results for stochastic switching characteristics of (c) Mo-MCM and (d) Ru-MCM. While Mo-MCM exhibited stochastic switching characteristics under a weak switching condition, Ru-MCM showed almost deterministic switching characteristics under a similar switching condition.

Our proposed HE-HW goes beyond merely generating encryption keys; it can also function as analog memory, thereby enabling in-memory computing between encryption keys and input data. This analog memory capability allows for the storage of both encrypted and decrypted input data and facilitates more efficient execution of vector matrix multiplication.

In summary, our device is not only capable of random bit generation but also serves as analog memory. Both functions are essential components of the HE-HW.

We elaborated on this aspect in the revised manuscript as follows.

Revised Manuscript (page 4):

“Although the cluster formation process is stochastic, once it is formed, its size can be accurately controlled by a compliance current, enabling analog conductance state programming. **This analog memory capability allows for the storage of both encrypted and decrypted input data and facilitates more efficient execution of VMM.** We confirm its digital random number generation and analog conductance programming capabilities from an integrated 1-Transistor-1-Memristor (1T1M) array device. **Then, we demonstrate the working of core operations experimentally and suggest the entire**

process of performing homomorphic encryption using the in-memory computing of hardware.”

< References in this answer >

1. Soto, J. in *Proceedings of the 22nd national information systems security conference*.
2. Kim, G. et al. *Nature Communications*. 12, 2906 (2021)
3. Woo, Kyung Seok, et al. *Advanced Intelligent Systems* 3.7 (2021)

Comment #3: “The distinctive feature of Mo-MCM is that its set switching is stochastic, but once it is set-switched, it shows stable analog states. If switching is stochastic, deterministic switching, e.g., weight update, would be costly to achieve. It could be very impractical for analog in-memory computing.”

We sincerely appreciate your thoughtful comment. In this revision, we have updated Fig. 1h with adding the stochastic switching results for intermediate pulse conditions (10 μ s and 30 μ s) to the existing data (stochastic switching of 1 μ s, and deterministic switching of 50 μ s). We hope the revised figure to demonstrate the stochastic to deterministic switching transition more clearly.

Here, in the 50 μ s condition, LRS represents the lowest conductance state among the deterministic switch modes. When the device is utilized as analog memory, higher conductance states are used by stronger pulse conditions, and more detailed results can be seen in Fig. 3.

Revised Figure (Fig. 1h):

Fig 1. h) Set switching probability according to the pulse conditions. As the set pulse width increases, both switching probability (P_{switch}) and the programmed conductance increase.

Revised Manuscript (page 6-7):

“The distinctive feature of Mo-MCM is that its set switching is stochastic, but once it is set-switched, it shows stable analog states. **Fig. 1h** show the transition behavior from the stochastic switching to deterministic switching. Here, the set-switching pulse widths were changed from 1 μ s to 50 μ s at 4 V. At a weak set condition, the device stochastically set-switched, so it randomly remained in the HRS even after receiving the set-switching pulse. The switching probability was increased as the pulse width increased, and eventually, a stable set- and reset-switching was obtained. Also, the conductance of the LRS was gradually increased even during stochastic switching, suggesting the analog state programming capability of Mo-MCM. (The potentiation characteristics under 1 μ s to 50 μ s at 4 V are shown in Supplementary Fig. S3.) The deterministic switching by 50 μ s at 4 V corresponds to the lowest conductance state, and more higher conductance states can be obtained, which is demonstrated in detail later.”

Revised Figure Caption (page 14):

“**Fig. 1.** Analysis of switching mechanism and electrical characteristics of Pt/Ta₂O₅/Mo memristor (Mo-MCM). a) Schematic images for the repeated formation and dissolution of cation nanoclusters

generated from Mo active electrodes in Mo-MCM. b) A top-view optical microscopy image of a single Mo-MCM. c) The resistive switching I-V curve of Mo-MCM. d) A time-dependent current-voltage measurement for Mo-MCM. e-f) $\tau_d - 1/V$ plot results for Mo-MCM and Pt/Ta₂O₅/Ru memristor (Ru-MCM) with W_0 obtained from the slope. g) A graph comparing the cluster nucleus size differences based on W_0 in Mo-MCM and Ru-MCM. h) Set switching probability according to the pulse conditions. As the set pulse width increases, both switching probability (P_{switch}) and the programmed conductance increase. ”

Comment #4: “In Fig. 1e and Fig. 1f, the authors obtain the data from only ten devices. I am wondering whether ten devices are enough to measure a sufficiently accurate τ_d - $1/V$ curve, although ten devices can also reflect the differences in trends to some extent.”

In response to the reviewer's comments, we collected data additionally from 30 devices (Total 180 cycles for single device) and updated the figures in Fig. 1e-f. During this measurement, we also used wider voltage ranges (Mo-MCM: 2.0 ~ 2.5 V, Ru-MCM: 4.0 ~ 4.5 V) to obtain more reliable fitting results. As a result, the nucleation barrier energy for both Mo-MCM and Ru-MCM was slightly changed (from 0.71 eV to 0.76 eV for Mo-MCM and from 0.97 eV to 0.99 eV for Ru-MCM). However, they have not affected our original claims.

Revised Figure (Fig. 1e-f):

Revised Manuscript (page 6):

“Fig. 1e and Fig. 1f are τ_d - $1/V$ plot results for Mo-MCM and Ru-MCM, respectively. Each data point on the plot were obtained from 30 devices (total 180 cycles for single device) with measurement range of 2.0 ~ 2.5 V for Mo-MCM, and 4.0 ~ 4.5 V for Ru-MCM.”

In addition, we have added the time-dependent current-voltage measurement results for Ru-MCM to the supplementary information in Fig. S2 for comparison with the Mo-MCM results in Fig. 1d.

Revised Supplementary Information (page 4):

Fig. S2. A time-dependent current-voltage measurement for Ru-MCM. The τ_d of Ru-MCM was shorter than Mo-MCM, suggesting the higher W_0 in Ru-MCM.

Revised Manuscript (page 6):

“**Fig. 1d** shows one of the results of the τ_d measurement in Mo-MCM, where the black line represents applied voltage, V , and the red line represents current response, I_{out} . The current response after a single pulse is saturated through a section that increases linearly. Here, τ_d is a time when the output current reaches the saturation value at a given applied voltage. (The τ_d measurement for Ru-MCM can be found at Supplementary **Fig. S2.**)”

Comment #5 “According to Fig. 1i, the R ratio of the proposed memristor is only 2~3. Could this low R ratio result in the accuracy loss? Furthermore, the average preservation rate is about 90%. Please explain the sources of the error and how to solve this problem.”

The conductance state shown Fig. 1i was the lowest conductance state, and the maximum on/off ratio can be as high as 100. The intention behind the original Fig. 1h-i was to show the difference between the stochastic switching and deterministic switching. In this revision, we revised the manuscript to explain the on/off ratio is the lowest conductance state and it can be higher, as demonstrated in Fig. 3c.

In the revised manuscript, we have supplemented the explanation for this part as follows.

Revised Manuscript (page 7):

“Also, the conductance of the LRS was gradually increased even during stochastic switching, suggesting the analog state programming capability of Mo-MCM. (The potentiation characteristics under 1 μ s to 50 μ s at 4 V are shown in Supplementary Fig. S3.) The deterministic switching by 50 μ s at 4 V corresponds to the lowest conductance state, and more higher conductance states can be obtained, which is demonstrated in detail later. Additionally, the endurance of 10^6 cycles was confirmed, making it sufficiently usable for HE-HW applications. (Endurance data can be found in Supplementary Fig. S4.)”

The average preservation rate is determined by the dispersion of the analog states. As shown in Fig. 3c, the interquartile range (IQR) error values for each state increase as the programmed conductance increases. In VMM, higher conductance states have a significant impact on the overall multiplication value, and thus, variations in high conductance states are identified as the cause of errors.

We believe that these variations can be overcome through process optimization via high-end CMOS processing or through advanced programming schemes such as Incremental Step Pulse Programming (ISPP). We have supplemented these discussion as follows.

Revised Manuscript (page 10):

“Data consistency comparisons are presented in Fig. 5b. The average preservation rate of addition in the data for Key A and Key B is denoted as 91.2% and 94.7%, respectively. Here, errors arise from variations in the analog states, particularly those in the higher conductance states, which significantly impact the VMM outputs. To enhance the preservation rate, achieving more uniform analog states is crucial. This can be accomplished through the use of a more advanced CMOS process or by implementing sophisticated programming schemes.^{45,46}”

Comment #6 “Relatively low performance in terms of switching speed under a practical voltage range (several microseconds under a few volts).”

As the reviewer correctly pointed out, the switching speed is slow, which is an inherent characteristic of metallic cluster type memristors. Below figures show the switching speeds of MCMs shown in other papers.

Reference graph: (Left) Ye, Fan, et al. *Advanced Materials* 35.37 (2023). (Right) Song, Young Geun, et al. *Advanced Science* 9.4 (2022): 2103484.

However, considering that our device is an encryption component and does not require repetitive programming operations, we believe that this drawback of slow switching speed can be less critical.

We added our discussion on the speed issue in the conclusion part as follows.

Revised Manuscript (page 11):

“Prior research has predominantly focused on integrating memristors with varying characteristics to develop sophisticated hardware systems. Our study sets itself apart by proposing a method to exploit the multiple inherent properties of a single memristor in complex hardware setups. We believe this research will establish a foundation for the wider adoption of memristors, which display a range of diverse and compelling phenomena, in various emerging hardware fields.

Nevertheless, the proposed devices and systems have the following potential improvements. Switching speed of Mo-MCM ranges from 1 µs for stochastic switching to 50 µs for deterministic switching, which is consistent with typical speeds reported in other literature. Therefore, improving the switching speed appears to be a challenging task. While this speed may be slow for dynamic memory applications, it does not significantly hinder its functionality in HE-HW, where stochastic switching and analog programming is advantageous.”

Comment #7 “Lack of the very important endurance characteristics with only about 1000 switching operations. This is critical for both random number generation and weight update.”

We thank for the comment. The results from the original figures with 10^3 cycles were intended to demonstrate the difference between stochastic and deterministic switching in our device, not to argue the device’s endurance. Based on the reviewer’s comments, we conducted endurance test and added its results in Fig. S4 in the supplement information. The endurance was as high as 10^6 cycles under the pulses of 4V, 50 μ s for set switching, and -4V, 1 μ s for reset switching.

Revised Supplementary Information (page 6):

Fig. S4. a) Endurance data for stochastic SET switching and analog computing with 10^4 cycles in a linear scale in two pulse schemes: (left) stochastic switching with 4V, $\sim 1 \mu$ s pulses, (right) deterministic switching with 4 V, 50 μ s pulses. Reset pulses were fixed to -4V, $\sim 100 \mu$ s. b) Endurance of the deterministic switching in a log scale for 10^6 cycles.

Also, we will make the following modifications to the manuscript accordingly.

Revised Manuscript (page 7):

“The deterministic switching by 50 μ s at 4 V corresponds to the lowest conductance state, and more higher conductance states can be obtained, which is demonstrated in detail later. **Additionally, the endurance of 10^6 cycles was confirmed, making it sufficiently usable for HE-HW applications. (Endurance data can be found in Supplementary Fig. S4.)**”

Comment #8 “The encryption and decryption key cannot be accessible from outside to ensure security. In the proposed hardware implementation, how to ensure that these keys are not accessible from the outside?”

Please refer to our response to Comment #1. In summary, the encryption key generated by the stochastic SET switching characteristic of the device will be stored within the memristive crossbar array. Also, its inverse matrix, as illustrated in Fig. S8 in the revised SI (it was Fig. S5 in the original SI), can be converted into voltages according to the given circuit, which then become new inputs to the memristive crossbar array without the need for a readout operation in the intermediate process. All processes are performed in-memory, and therefore, data reading is fundamentally impossible without a separate read instruction.

Comment #9 “Typo: In lines 143 and 144, Fig. 1g should be Fig. 1i; P5: a mean to estimate  a means to estimate; P7: a LRS ratio  an LRS ratio.”

Thank you very much for the detailed review of our manuscript. We have corrected the typos pointed out by the reviewer, and additionally, we have thoroughly reviewed the entire manuscript.

[Response to Reviewer #2]

We thank the reviewer for a thoughtful review and comments. We hope that our revision responds well to the comments.

Comment #1: “Details on the dynamic information (pulse width and voltage interval) of I-V hysteresis curve measurement in Fig.1c is recommended to include.”

We sincerely appreciate your thoughtful comment. The I-V curve in Fig. 1c was obtained under a DC sweep mode, while Fig. 1h were obtained by a pulse mode. We clarified it in the revised manuscript as follows.

Revised Manuscript (page 5):

“The cross-section area is $5 \times 5 \mu\text{m}^2$. A detailed device fabrication process can be found in the Experimental Section. The **direct current (DC)** resistive switching current-voltage (I-V) curve of Mo-MCM is shown in **Fig. 1c.**”

Also, we added *Electrical characterization* part in Methods section.

Revised Manuscript (page 12):

“Electrical characterization

The electrical characterization in Fig. 1 was performed using Keithley 4200A-SCS and a hot chuck controller (MST-1000H) was used for the ambient temperature control. During the electrical measurement, the top electrode was biased while the bottom electrode was grounded. The electrical characterization of 1T1M devices was performed using custom-made measuring equipment based on ArC ONE from ArC instrument©.”

Comment #2: “Reporting the read current (or conductance) vs. # of pulse such as Fig.10 and Fig.11 of [Marinella et al. IEEE Journal on Emerging and Selected Topics in Circuits and Systems 8.1 (2018): 86-101] is recommended as a supporting information for readers in the compact model area. This will also reveal whether the stochastic-switching is attributed to mainly the physical mechanism hypothesized by Fig.1g or also in part by significant nonlinearity in switching.”

We thank for the suggestion. Before we investigate the read current vs # of pulse behavior (we call it a potentiation behavior), we updated Fig. 1h to show switching transition from stochastic switching to deterministic switching as follows.

Revised Figure (Fig. 1h):

Based on this result, we measured 10 cycles potentiation behavior using stochastic switching condition (4 V / 1 μ s) and deterministic switching condition (4 V / 50 μ s), whose results are shown below. In the stochastic switching condition (left panel), variation was observed during at the first set switching due to the stochastic nature of the switching. However, once it set switched, the subsequent pulses increased the conductance gradually as a typical potentiation behavior. Consequently, there was a variation at the first potentiation step. In the deterministic switching condition (right panel), typical potentiation behavior was observed.

Based on the reviewer’s comments, we will add the results to the supplementary information in Fig. S3.

Revised Supplementary Information (page 5):

Fig. S3. 10 cycles of potentiation characteristics for stochastic switching condition (4 V / 1 μ s, left panel) and deterministic switching condition (4 V / 50 μ s, right panel) are shown. there is variation during the initial switching, which is related to the probability of stochastic switching occurring. Once it has switched, the device exhibits typical potentiation behavior. In the deterministic switching condition, typical potentiation characteristics were observed. The saturation current at 20 pulses was about 4 nA in the stochastic switching condition and about 40 nA in the deterministic switching condition. This difference originated from the difference pulse widths for two conditions.

Revised Manuscript (page 7):

“The switching probability was increased as the pulse width increased, and eventually, a stable set-and reset-switching was obtained. Also, the conductance of the LRS was gradually increased even during stochastic switching, suggesting the analog state programming capability of Mo-MCM. (The potentiation characteristics under 1 μ s to 50 μ s at 4 V are shown in Supplementary Fig. S3.)”

Comment #3: “Authors are also recommended to provide Ru-memristor’s raw data equivalent to Fig. 1h and the read current vs. # of pulse plot suggested in Comment#1 as a supporting information. These raw data will be more convincing to readers in addition to the hypothesis in Fig. 1g.”

We thank for the suggestion. We added Ru-MCM characteristics and compared Mo-MCM and Ru-MCM switching characteristics in Fig. S1 in the supplementary information as follows.

Revised Supplementary Information (page 3):

Fig. S1. Electrical characteristics of Ru-ECM (Pt/Ta₂O₅/Ru) memristors. a) I-V characteristic of a Ru-ECM memristor with 5 repetitions, which shows stable and repetitive switching behavior. b) Schematic image for a vertical structure of Ru-ECM memristors. The bottom Pt electrode was deposited using E-beam evaporation with a thickness of 40 nm, followed by a deposition of 20 nm of Ta₂O₅ using DC reactive sputtering with a Ta target. The top Ru electrode was deposited with a thickness of 40 nm using DC sputtering. c-d) Comparison results for stochastic switching characteristics of (c) Mo-MCM and (d) Ru-MCM. While Mo-MCM exhibited stochastic switching characteristics under a weak switching condition, Ru-MCM showed almost deterministic switching characteristics under a similar switching condition.

Comment #4: “In Fig.3c, 8th state shows 200 nA, which even exceeds the SET state current (w/ 0.5V read voltage) in Fig.1c (30 nA). It will be informative to mention the endurance with a simple statement (e.g., 500 cycles, 100 cycles, etc.) for readers to know how much it deteriorates compared to limited range endurance test (>1000 cycles in Fig.1i).”

We appreciate your thoughtful comment.

For memory reading, we used 1.0 V. In Fig. 1c, the read current of the LRS was about 200 nA at 1.0 V, well matching with the 8th state (the highest conductance state) in Fig. 3c.

We have specified the read voltage in the pulse switching results and indicated the current levels on the I-V curves in Fig. 1c to demonstrate the consistency of the results.

Revised Figure (Fig. 1c):

Additionally, we included endurance results up to 10^4 cycles in a linear scale and 10^6 in a log scale in the revised supplementary information as below.

Revised Supplementary Information (page 6):

Fig. S4. a) Endurance data for stochastic SET switching and analog computing with 10^4 cycles in a linear scale in two pulse schemes: (left) stochastic switching with 4V, $\sim 1 \mu\text{s}$ pulses, (right) deterministic switching with 4 V, $50 \mu\text{s}$ pulses. Reset pulses were fixed to -4V, $\sim 100 \mu\text{s}$. b) Endurance of the deterministic switching in a log scale for 10^6 cycles.

Also, we will make the following modifications to the manuscript accordingly.

Revised Manuscript (page 7):

“Additionally, the endurance of 10^6 cycles was confirmed, making it sufficiently usable for HE-HW applications. (Endurance data can be found in Supplementary Fig. S4.)”

Comment #5: “The difference between $5 \times 5 \mu\text{m}^2$ memristor (single), $10 \times 10 \mu\text{m}^2$ memristors (in 16×24 array)? According to Fig.2 and Fig.3, it seems that the memristor’s conductance doesn’t change much even with larger device cross section. If the underlying physics is bulk-switching, instead of filamentary-switching, an explanation on this would be helpful.”

We apologize for our mistake and the resulting confusion. The scale bar in the original version was incorrect. It should be $250 \mu\text{m}$ in Fig. 1a, right panel, and $10 \mu\text{m}$ in Fig. 1b. (Previously, they were twice as large.) We sincerely appreciate the reviewer’s meticulous review, which allowed us to identify this error. In all devices, we have made considerable efforts to maintain a consistent line width to ensure that the behavior observed in single devices is accurately replicated in the array.

In response to the reviewer’s feedback, we fixed the figure caption of Fig. 2 as follows:

Revised Figure Caption (page 15):

Fig. 2. 1T1M array integration of Mo-MCM and its evaluation on uniformity. a) A photograph of the 1T1M device on a glass substrate and an optical microscope image of the cells at the core area. Scale bars indicate 1 cm (left) and $250 \mu\text{m}$ (right). b) A magnified optical microscope image of single 1T1M cell. A scale bar indicates $10 \mu\text{m}$. c) A schematic image of the cross-section of the 1T1M device. d) Heatmaps for 4 iterations of random number generation from stochastic set-switching in 5×5 sub-arrays. e) Current distribution of LRS and HRS on each iteration. f) Illustration of the process for calculating LRS ratio by stochastic set-switching. g) A graph for the ratio of LRS after stochastic set-switching as a function of the cycling number. h) A graph for the stochastic set-switching uniformity of 9 cells across 100 cycles.

Also, we clarified the Mo-MCM device area size in the array in the revised manuscript as follows.

Revised Manuscript (page 7):

“**Fig. 2a** shows a photograph of the integrated die on a glass substrate (left panel), whose size is 3×3 cm^2 , and the optical microscopy image of the core area (right panel), where each cell has a size of 70×65 μm^2 , including a Mo-MCM of 5×5 μm^2 .”

[Response to Reviewer #3]

Comment #1: “The authors claimed that “the metallic cluster formation process is intrinsically stochastic due to the slow migration characteristics of the Mo atoms on redox reactions, resulting in a wide switching voltage variation”. This can mislead the readers, as memristors based on Ag and Cu are also known to exhibit significant variation.”

We sincerely appreciate your thoughtful comment. As the reviewer noted, all types of cluster-based memristors exhibit a certain degree of stochastic characteristics. Our Mo-MCM, by virtue of its material properties, shows more pronounced stochastic behavior.

In the revision, we have clarified this to avoid any potential confusion among readers.

Revised Manuscript (page 4):

“In Mo-MCM, we identified that the set switching process is highly stochastic due to the intrinsic high interfacial energy of Mo, which complicates and randomizes the ionization process from the active electrode.^{35,36} This results in a wide switching voltage variation, enabling its use in random number generation.”

Comment #2: “What makes the device to switch in a cluster-type rather than a filamentary-type? The authors cited ref [27] for the concept of Ru-MCM. In Ref [27], the switching mechanisms were modulated by controlling the I_{cc} . Can the authors show similar behaviors in both Ru-MCM and Mo-MCM? Please explain more about the cluster-type switching.”

We thank the reviewer for bringing this point to our attention. The difference between filament-type and cluster-type lies in the nature of the conducting path in the LRS; filament type is characterized by a continuous wire-like connection, whereas cluster type involves a discontinuous connection of clusters. The LRS panel in Fig. 1a illustrates it.

Filament-type typically exhibits set switching at very low voltages, high LRS current and on/off ratio, and a drastic set switching behavior. In contrast, cluster type memristors have a relatively higher set voltage, lower LRS current, and demonstrate analog set switching characteristics.

We elaborated on this point in the revised manuscript as follows.

Revised Manuscript (page 4-5):

“In this device, the switching mechanism is attributed to the repeated formation and dissolution of Ru nanoclusters generated from active electrodes. Unlike filament type switching (or electrochemical metallization), where continuous filament bridges form, the clusters in this device are discontinuous. As a result, electrical conduction relies on a hopping mechanism, allowing operation in a low current range. Additionally, the gradual formation of clusters makes this device well-suited for analog switching. However, the formation process of Ru nanoclusters is insufficiently stochastic to be used for the encryption key generator. Therefore, we endeavored to comprehend the nanocluster formation process and explored new mobile species that inherently introduce stochasticity to this process. As a result, we identified Mo-MCM, where Mo, as a mobile species, can provide such functionalities.”

As the reviewer pointed out, the clusters are known to evolve to the filament at a higher current. We could observe the filamentary-type switching at a high compliance current ($>10^{-5}$ A) as below in Fig. R1. We believe that these results can enhance the credibility of our claims.

Fig. R1 The resistive switching I-V curves of Mo-MCM at the compliance current of sub-mA. During the set/reset process, rapid threshold switching characteristics similar to electrochemical metallization (ECM) can be observed. This suggests that the initially formed clusters evolve into continuous filaments. These results are similar to those shown in Ref. 34 (it was Ref. 27 in the original manuscript) for Ru-MCM, indicating that the switching characteristics of Ru-MCM and Mo-MCM are very similar.

Comment #3: “Fig. 1a does not show the switching based on Ru nanoclusters.”

We are sorry for the confusion. In this revision, we have updated the manuscript to ensure that Figure 1 now accurately illustrates the operating principles of Mo-MCM.

Revised Manuscript (page 5):

“Fig. 1a illustrates the suggested switching mechanism of the Mo-MCM.”

Comment #4: “The authors should make another plot for $V_{set} - V_{HRS}$ to show there is no correlation.”

To our understanding, it seems that reviewer suggested a demonstration that there is no correlation between set voltage and reset voltage according to the repetitive switching behavior. As shown in the graph below, while maintaining the scheme of the reset pulse, varying the scheme of set pulse results in changes in the switching probability along with the degree of analog programming. As evident from the HRS value remains constant, indicating no correlation between set voltage and reset voltage.

We chose the condition of 4 V, 1 μ s pulse scheme as the stochastic switching pulse condition because the LRS/HRS ratio at this condition achieves nearly 50%. This is closely related to the NIST test, where achieving a 50% probability condition is considered the most important aspect among the conditions for randomness. Therefore, when generating encryption keys, using the pulse conditions that achieve approximately 50% probability is ideal.

Following the reviewer's comment, we will replace the Fig. 1h of the manuscript.

Revised Figure (Fig. 1h):

Fig 1. h) Set switching probability according to the pulse conditions. As the set pulse width increases, both switching probability (P_{switch}) and the programmed conductance increase.

Revised Manuscript (page 6):

“The distinctive feature of Mo-MCM is that its set switching is stochastic, but once it is set-switched,

it shows stable analog states. **Fig. 1h** show the transition behavior from the stochastic switching to deterministic switching. Here, the set-switching pulse widths were changed from 1 μs to 50 μs at 4 V. At a weak set condition, the device stochastically set-switched, so it randomly remained in the HRS even after receiving the set-switching pulse. The switching probability was increased as the pulse width increased, and eventually, a stable set- and reset-switching was obtained. Also, the conductance of the LRS was gradually increased even during stochastic switching, suggesting the analog state programming capability of Mo-MCM. (The potentiation characteristics under 1 μs to 50 μs at 4 V are shown in Supplementary Fig. S3.) The deterministic switching by 50 μs at 4 V corresponds to the lowest conductance state, and more higher conductance states can be obtained, which is demonstrated in detail later.”

Revised Figure Caption (page 14):

“**Fig. 1. Analysis of switching mechanism and electrical characteristics of Pt/Ta₂O₅/Mo memristor (Mo-MCM).** a) Schematic images for the repeated formation and dissolution of cation nanoclusters generated from Mo active electrodes in Mo-MCM. b) A top-view optical microscopy image of a single Mo-MCM. c) The resistive switching I-V curve of Mo-MCM. d) A time-dependent current-voltage measurement for Mo-MCM. e-f) $\tau_d - 1/V$ plot results for Mo-MCM and Pt/Ta₂O₅/Ru memristor (Ru-MCM) with W_0 obtained from the slope. g) A graph comparing the cluster nucleus size differences based on W_0 in Mo-MCM and Ru-MCM. h) Set switching probability according to the pulse conditions. As the set pulse width increases, both switching probability (P_{switch}) and the programmed conductance increase. ”

Comment #5: “Please provide more information on the NIST test in the main text or supplementary. How many devices and bits were used?”

Under the condition of 1,000 bits - 100 sequences, we conducted NIST randomness suit tests for stochastic SET behavior of Mo-MCM. During the NIST tests, the 'Overlapping template matching' test, which requires a minimum of 10^6 bits of data per sequence, was excluded due to the difficulty of measuring according to the device specifications. Therefore, we concluded that it would be difficult to claim our device as a true random number generator (TRNG) but as a pseudo-random number generator (PRNG).

Following the reviewer's suggestion, we will update Table S1 of supplementary information with our NIST test condition and the proportion of passes for each test as follows.

Revised Supplementary Information (page 2):

Table S1. NIST **randomness suit** test results for evaluating random number generation performance required for homomorphic encryption. **Each test for Mo-MCM was conducted under the condition of 1,000 bits - 100 sequences.**

Test #	Test name	P-value (Mo-MCM)	Proportion (100 seq.)	Result (Mo-MCM)
1	Frequency (Monobit)	0.011412036	0.98	Pass
2	Frequency within a block	0.550717949	0.99	Pass
3	Runs	0.532759772	1.00	Pass
4	Longest run of ones in a block	0.348025369	0.98	Pass
5	Binary matrix rank	0.015710521	0.98	Pass
6	Discrete Fourier transform (Spectral)	0.771670504	0.99	Pass
7	Non-overlapping template matching	1.00	1.00	Pass
8	Maurer’s universal	0.988785497	1.00	Pass
9	Linear complexity	0.999995031	1.00	Pass
10	Serial	0.761854261	0.99	Pass
11	Approximate entropy	0.999520079	0.99	Pass
12	Cumulative sums	0.017345555	0.97	Pass
13	Random excursions	0.950167133	0.98	Pass
14	Random excursions variant	0.910979293	0.99	Pass

Revised Manuscript (page 7):

“The stochastic set-switching characteristic can be used for the random number generator by defining the LRS (1) and HRS (0) states for digital output. **To ensure its randomness, we conducted the NIST**

randomness suite tests using 1,000 bits – 100 sequences condition^{12,13,44}. At the optimized set condition (4V, 1 μ s), the output data string passed all of the 14 NIST test criteria. This test was performed at highly controlled conditions to demonstrate a Pseudo Random Number Generator (PRNG) capability so that our device can certainly be used as a reliable random number generator, which is sufficient for utilization in homomorphic encryption.”

Comment #6: “Can the authors provide more discussion on HE-HW? For example, would performing multiple additively HE decrease the preservation rate? If so, what would limit the performance of HE? Is it also possible to perform multiplicatively HE?”

Thank you for the suggestion. In this study, we demonstrated additively homomorphic encryption as a proof of concept. We acknowledge the reviewer’s suggestion that multiplicatively HE represent more advanced forms of HE (i.e. Fully HE), and it is clear that these approaches could enhance usability as a security device. These advanced systems require complex functionality and precise operations, which makes them challenging to address fully in this study. Therefore, we wish to conclude our research by noting the potential of these methods to improve the device functionalities, acknowledging that they warrant further exploration in future work.

We discussed this at the discussion section as below.

Revised Manuscript (page 11):

“Next, we demonstrated additively HE with a preservation rate of about 90%, which needs to verify whether it can be said to be Fully HE through multiplicatively HE demonstration. In order to demonstrate multiplicatively HE, which requires complex functionality and precise operations, the device's analog states uniformity must be improved than now. These strategies will require further in-depth research to fully realize their potential.”

REVIEWERS' COMMENTS

Reviewer #1 (Remarks to the Author):

Thank the authors for their thorough revisions and responses to the problems I previously mentioned. I appreciate the authors' effort and their responses have resolved all of my questions and concerns. I have no further comments or additional questions.

Reviewer #2 (Remarks to the Author):

Authors addressed my comments satisfactorily. Now it is acceptable for publication.

Reviewer #3 (Remarks to the Author):

The authors have addressed all my concerns. I recommend it for publication.